# Purification, Characterization and Bioactivity of Different Molecular-Weight Fractions of Polysaccharide Extracted from Litchi Pulp

**DOI:** 10.3390/foods12010194

**Published:** 2023-01-01

**Authors:** Xiaoqin Zou, Jiaxin Cai, Jiaxi Xiao, Mingwei Zhang, Xuchao Jia, Lihong Dong, Kun Hu, Yang Yi, Ruifen Zhang, Fei Huang

**Affiliations:** 1Guangdong Key Laboratory of Agricultural Products Processing, Key Laboratory of Functional Foods, Ministry of Agriculture and Rural Affairs, Sericultural & Agri-Food Research Institute Guangdong Academy of Agricultural Sciences, Guangzhou 510640, China; 2Hubei Key Laboratory for Processing and Transformation of Agricultural Products, College of Food Science and Engineering, Wuhan Polytechnic University, Wuhan 430023, China; 3Food Science School, Guangdong Pharmaceutical University, Zhongshan 528458, China

**Keywords:** litchi pulp polysaccharide, ultrafiltration membrane, physicochemical properties, immunomodulatory activity, prebiotic activity

## Abstract

Litchi polysaccharides are a kind of macromolecular polymers with various biological activities and a wide range of molecular weights. In this study, two separate fractions, with average molecular weights of 378.67 kDa (67.33%) and 16.96 kDa (6.95%), which were referred to as LP1 and LP2, respectively, were separated using an ultrafiltration membrane. Their physicochemical properties, and immunomodulatory and prebiotic activity were compared. The results revealed that LP2 contained more neutral sugar, arabinose, galactose and rhamnose, but less uronic acid, protein, mannose and glucose than LP1. Compared with LP1, LP2 possessed higher solubility and lower apparent viscosity. LP2 exhibited stronger stimulation on macrophage secretion of NO, TNF-α and IL-6, as well as better proliferation of *Lactobacillus plantarum*, *Leuconostoc mesenteroides*, *Lactobacillus casei* and *Bifidobacterium adolescentis*. These results suggest that an ultrafiltration membrane might be used to prepare a highly-active polysaccharide fraction from litchi pulp that may be used for food or drug development.

## 1. Introduction

Litchi (*Litchi chinensis* Sonn.) is a traditional fruit extensively grown in subtropical regions for its medicinal and food properties [1]. Litchi pulp has long been used in traditional Chinese medicine as a remedy for diabetes, obesity, diarrhea, and stomach ulcers [2,3]. According to preliminary research, polysaccharide is the principal active ingredient in litchi pulp, and has a multitude of health-promoting properties, including antioxidant, immunomodulatory, and anticancer effects [4,5]. The activity of litchi polysaccharide is strongly correlated with its molecular weight, monosaccharide composition, and glycosidic linkages. Nevertheless, the litchi polysaccharide structure is quite complex. For instance, Huang [6] discovered that the molecular weights of litchi polysaccharides ranged from 105.88 to 986.47 kDa, whereas 2400 kDa and 47.2 kDa were detected from the research of Yang [7] and Jing [8], respectively. Furthermore, Hu [9] found that the backbone of litchi polysaccharide consisted of →3)-β-L-Rha-(1→, →4)-α-D-Xyl-(1→ and →4)-β-D-Glu-(1→, but Yang [7] claimed that it was composed of →4)-α-D-GalpA6Me-(1→ and →6)-β-D-Galp-(1→. In current studies, an anion-exchange DEAE cellulose column and Sephadex and Sephacryl gel column are commonly used for the separation and purification of litchi polysaccharides; however, they are difficult to apply in food and pharmaceutical industries due to their high manufacturing cost and poor efficiency [10]. Additionally, the widespread use of organic solvents and ionic liquids in many purification techniques easily raises security concerns due to residual chemical reagents [11]. Hence, exploring a low-cost, high-efficiency approach for litchi polysaccharide purification is required.

Ultrafiltration membrane separation is a process in which macromolecules are selectively separated from a solution by using the principle of mechanical sieving with pressure as the driving force [12]. Because of its advantages of environmentally operation, high efficiency, and cheap cost, membrane separation technology is a revolutionary separation technique that has been used in many areas. For instance, Tsou et al. [13] quickly and effectively produced anti-adipogenic active components by separating soy proteins using ultrafiltration membranes with various molecular weight cut-offs. Liu et al. [14] isolated polysaccharides from *Tricholoma lobayense* using an ultrafiltration membrane with a molecular weight cut-off of 10 kDa, achieving the separation of *Tricholoma lobayense* polysaccharides in large quantities. This demonstrates that ultrafiltration membrane separation is a clean, effective method that can be employed on a wide scale. However, there is no information available regarding its use in litchi polysaccharides.

Furthermore, previous researches have demonstrated that litchi polysaccharides possess immunomodulatory [15], antioxidant [16], antiproliferative [5], and prebiotic activity [17]. Among them, the immunological activity of polysaccharides is the most widely reported [18]. Some polysaccharides have been used as immunomodulators in clinical cancer therapy, such as *Lentinus edodes* and *Ganoderma lucidum* [19]. Thus, macrophages were employed to track the activity of litchi polysaccharide following ultrafiltration membrane purification in this study. In addition, when investigating the mechanism of polysaccharide activity, it has been discovered that polysaccharide primarily regulates the gut microbiota to perform these effects. For example, *Ziziphus Jujuba* cv. *Pozao* polysaccharide could significantly promote splenic lymphocyte proliferation in ICR mice and positively regulate the gut microbiota as indicated by the enriched microbiota diversity [20]. Similar results were found with *Lycium Barbarum* polysaccharide (LBP) [21]; the innate immune system can be modulated by LBP, which can also improve the intestinal microbiota and increase amounts of beneficial bacteria. Hence, evaluation of the prebiotic activity of litchi polysaccharide is crucial for determining its likely impact on health. As a result, the immunomodulatory and prebiotic activities of litchi polysaccharide fractions separated by ultrafiltration membranes were further compared and analyzed.

This study aims to (1) establish a quick and effective method for the preparation of litchi polysaccharide; (2) analyze the physicochemical and structural characteristics of polysaccharides with different molecular weights; and (3) expound on the relationship between the polysaccharide structure and its immunomodulatory and prebiotic activity. Consequently, establishing an efficient preparation technology for highly-active litchi polysaccharide is expected, which will improve the possibility of its exploitation and utilization in the pharmaceutical industry.

## 2. Materials and Methods

### 2.1. Materials and Chemicals

#### 2.1.1. Chemicals and Reagents

Fresh litchi fruits (cv. Huai-zhi) were provided by the Fruit Tree Research Institute of Guangdong Academy of Agricultural Sciences (Guangzhou, China). Rhamnose, arabinose, glucose, galactose, mannose, xylose, fucose and standard dextran (T-4: molecular mass, 4.4 × 10^3^ Da, T-10: 9.9 × 10^3^ Da, T-20: 2.14 × 10^4^ Da, T-40: 4.35 × 10^4^ Da, T-100: 1.24 × 10^5^ Da, T-200: 1.96 × 10^5^ Da, T-300: 2.77 × 10^5^ Da, and T-400: 4.01 × 10^5^ Da), lipopolysaccharide (LPS), penicillin and streptomycin were purchased from Sigma Chemical Co. (St. Louis, MO, USA). Dulbecco’s Modified Eagle Medium (DMEM) and fetal bovine serum (FBS) were supplied from Gibco Inc. (Grand Island, NY, USA). Enzyme-linked immunosorbent assay (ELISA) kits were obtained from NeoBioscience (Shenzhen, China). Fructooligosaccharides (DP 2–9) was acquired from Shanghai Ruiyong Biotechnology Co., Ltd. (Shanghai, China). Man-Rogosa-Sharpe (MRS) was supplied from Guangdong Huankai Microbial Technology Co., Ltd. (Guangzhou, China). All other reagents were analytical grade.

#### 2.1.2. Chemicals and Reagent

The mouse RAW 264.7 macrophages were purchased from Shanghai Institutes for Biological Science, CAS (Shanghai, China). Cells were cultured in RPMI-1640 medium containing 10% fetal calf serum, 100 U/mL penicillin and 100 μg/mL streptomycin at 37 °C and 5% CO_2_.

*Lactobacillus plantarum* GIM 1.380, *Leuconostoc mesenteroides* GIM 1.473, *Lactobacillus casei* GIM 1.411 and *Bifidobacterium adolescentis* ATCC 15,703 were acquired from Guangdong Microbial Culture Collection Center. The strains were stored in MRS broth containing 20% glycerol (*v*/*v*) at −80 °C until use. The bacteria were activated in accordance with the earlier techniques before testing [22].

### 2.2. Isolation and Purification of Polysaccharides

Fresh litchi fruits were dried using hot air drying at 65 °C. In order to remove the pigments, monosaccharides, and oligosaccharides, the dried pulps were steeped in 80% ethanol. The residues were filtered, then homogenized, before being extracted twice with distilled water (1:20, g/mL) at 95 °C for four hours. The extracts were filtered and concentrated to an appropriate volume. Subsequently, the concentrates were subjected to the Sevag reagent to remove free proteins [23]. After centrifugation (4000 rpm, 15 min), the deproteinated solution was precipitated with absolute ethanol (1:4, *v*/*v*). The precipitates were lyophilized to obtain crude litchi pulp polysaccharides (CLP).

In total, 100 mg of CLP was dissolved in 20 mL of distilled water. Ultrafiltration purification was carried out by ultrafiltration membrane (UFP-100-E-3MA, GE Healthcare Bio-sciences Co., Ltd., Piscataway, NJ, USA) with a molecular weight cut-off of 100 kDa, and the purified fractions were collected and lyophilized to obtain LP1 and LP2 with molecular weights greater than 100 kDa and less than 100 kDa, respectively. The polysaccharide yield was determined.
(1)$$\mathrm{The}\;\mathrm{yield}=\frac{m_{1}}{m_{2}}$$
wherein *m*_1_ and *m*_2_ are the masses of lyophilized purified fraction and lyophilized crude polysaccharide.

### 2.3. Physicochemical Properties of LPs

#### 2.3.1. Chemical Composition

The phenol-sulfuric acid method was used to determine the neutral sugar concentration, and glucose was modified as the standard [24]. Using galacturonic acid standards and the m-hydroxydiphenyl technique, the concentration of uronic acid was determined [25]. The Bradford assay was used to determine the protein content [26].

#### 2.3.2. Molecular Weights

By using an Acquity advanced polymer chromatography system (APC) fitted with a refractive index detector, the average molecular weights of the LPs were ascertained [22]. Acquity APC AQ 45, 200, and 450 columns (4.6 mm 150 mm, Waters Corp., Milford, MA, USA) were used to accomplish the separation. The column oven temperature was set at 35 ± 0.2 °C and the column was eluted with 50 mM Na_2_SO_4_ at a flow rate of 0.7 mL/min. Standard dextran including T-4 (molecular mass, 4.4 × 10^3^ Da), T-10 (9.9 × 10^3^ Da), T-20 (2.14 × 10^4^ Da), T-40 (4.35 × 10^4^ Da), T-100 (1.24 × 10^5^ Da), T-200 (1.96 × 10^5^ Da), T-300 (2.77 × 10^5^ Da), and T-400 (4.01 × 10^5^ Da) were used as molecular mass markers.

#### 2.3.3. Monosaccharide Composition

The monosaccharide composition of the LPs was determined by GC-MS according to our previous method with some modifications [15]. Briefly, 10 mg of each sample was hydrolyzed with 4.0 M trifluoracetic acid (2.0 mL) at 110 °C for 6 h, and then evaporated to dryness under reduced pressure at 50 °C. After repeatedly dissolving in methanol three times, the dried hydrolyzed samples were dissolved in 10 mg hydroxylamine hydrochloride, 1.0 mL pyridine and 1.0 mL acetic anhydride, in turn, for derivatization at 90 °C for 30 min. The samples, mixed with proper anhydrous Na_2_SO_4_, were filtered through 0.2 μm syringe filters. The final sample was analyzed by GC-MS, using a Shimadzu GC-2010 Plus system (Shimadzu, Kyoto, Japan) equipped with a HP-1701 column (30 × 0.25 m i.d. 0.33 μm). The temperature program was set as follows: the initial column temperature was 190 °C, increased to 230 °C at 2 °C/min, holding for 2 min, then to 240 °C at 5 °C/min, holding for 2 min. The detector temperature was 290 °C and the vaporizing chamber temperature was set at 260 °C. Several monosaccharides (arabinose, mannose, rhamnose, galactose, xylose, fucose and glucose) were used as the external standards to identify the composition of the polysaccharides.

#### 2.3.4. Solubility

According to the research of Huang et al. [27], 500 mg of each sample was kept in 5 mL of distilled water and swirled magnetically for 6 h. A centrifuge (3000 rpm, 10 min) was used to separate the precipitation from the supernatant. Following drying and weighting, the precipitation and supernatant were combined. The solubility of the samples was calculated as the weight of polysaccharide per milliliter of water.

#### 2.3.5. Apparent Viscosity

The concentration of the polysaccharide solution was 20 mg/mL. The AR1500EX rheometer (TA Instruments, New Castle, DE, USA) was used for apparent viscosity determination, and the temperature was kept at 25 °C. Configure the Spindle of CP52 (Shear Rate Constant = 2, Spindle Multiplier Constant = 9.83) [27].

#### 2.3.6. Fourier Transform-Infrared (FT-IR) Spectroscopy

For FT-IR analysis [28], 1 mm thick pellets of 2 mg polysaccharide samples and 100 mg potassium bromide (KBr) powder were formed. On a Nexus 5DXC FT-IR (Thermo Nicolet, Austin, TX, USA), FT-IR spectra were captured in the frequency range of 4000–400 cm^−1^ at 4 cm^−1^ resolution using 64 scans.

### 2.4. Immunomodulatory Activity of LPs

Preliminary results had showed that LP1 and LP2 had no toxic effect on macrophages in the concentration range of 0–400 μg/mL. Briefly, RAW264.7 macrophages (5 × 10^5^ cells/mL) were added into a 96-well culture plate at 100 μL/well. After incubating for an appropriate time (37 °C, 5% CO_2_), the unadhered cells were washed twice with PBS. Subsequently, 100 μL/well of the culture medium containing the polysaccharide solution (the final concentrations of LP1 and LP2: 50, 75 and 100 μg/mL) was added, the complete medium was used as the blank control, and 1 ng/mL LPS as the positive control. After 24 h of incubation, the supernatant was collected and NO, IL-6 and TNF-α secretion were determined according to the operating instructions of the mouse NO kit and IL-6 or TNF-α ELISA kit [6].

### 2.5. Prebiotic Activity of LPs

According to our prior research, we selected four lactic acid strains, such as *Lactobacillus plantarum*, *Leuconostoc mesenteroides*, *Lactobacillus casei* and *Bifidobacterium adolescentis*, to determine whether LP1 and LP2 exhibited probiotic activity [29]. To investigate the prebiotic activity of LPs in vitro, carbohydrate-free MRS broth containing 0.05% (*w*/*v*) L-cysteine was chosen as the baseline medium, the baseline MRS medium was used as the blank control, and the recognized prebiotic, fructooligosaccharide (FOS), was used as the positive control. The carbohydrates (LP1, LP2 and FOS) were filtered and sterilized before being added to the baseline MRS broth at a final concentration of 2.0% (*w*/*v*). To generate a concentration of probiotics with 1 × 10^6^ CFU/mL, the active lactic acid bacteria were injected into the MRS broth medium. The samples were then cultured in an anaerobic chamber for 48 h at 37 °C with 85% N_2_, 10% CO_2_, and 5% H_2_. After 0 and 48 h of incubation, the lactic acid strain counts of the samples were tallied. The procedures for the bacterial enumeration were followed. Using sterile 1% (*w*/*v*) peptone solution, one milliliter of the culture was transferred to sterile test tubes and serially diluted. Then, 100 L of each dilution was applied to the MRS agar plates, and anaerobic incubation was conducted at 37 °C for 48 h. The number of bacteria was counted and expressed as log CFU/mL. Following the methodology below, the rise in bacterial counts between 0 and 48 h was estimated.
Increase of bacterial number = log B − log A(2)
where A was the bacterial number at 0 h (CFU/mL) and B was the bacterial number after incubation for 48 h (CFU/mL).

### 2.6. Statistical Analysis

All the experiments were tested at least three times independently. The means ± standard deviation (SD) were used to express the data. Using SPSS 19.0 software, a one-way ANOVA and Student–Newman–Keuls test were used to determine the significance. The cutoff for significance was set at a *p*-value of 0.05.

## 3. Results and Discussion

### 3.1. Polysaccharide Purification

In order to explore the separation effect of the ultrafiltration membrane on crude litchi polysaccharide, APC was used to analyze the molecular weight distribution of crude polysaccharide and the separated litchi polysaccharide components; the results are shown in Figure 1. Obviously, the peak pattern of crude polysaccharide was widely distributed from 2–9 min, and there are some differences between the peak shapes of LP1 and LP2. The peak appearance time of LP1 is 2–8 min, while that of LP2 is 4–9 min, which is later than that of LP1, indicating that the Mw of LP2 is significantly smaller than that of LP1. It can be seen that the Mw of crude litchi polysaccharide is 297.75 kDa, and the Mw of two litchi polysaccharide fractions are 378.67 and 16.96 kDa, respectively. The findings indicated that screening litchi polysaccharides with significant molecular weight variations could be accomplished effectively using an ultrafiltration membrane.

### 3.2. Physicochemical Properties

#### 3.2.1. Basic Characterization

The chemical composition analysis of LP1 and LP2 is presented in Table 1. The yields of LP1 and LP2 were significantly different, coming in at 67.33% and 6.95%, respectively. The sugar contents of the LPs were over 80%, but LP1 possessed a higher uronic acid content, while LP2 was higher in neutral sugar content. Protein content was quite low in both LP1and LP2. Table 1 also shows the solubility and apparent viscosity variations between LP1 and LP2. The solubility of LP2 is twice that of LP1, while the viscosity of LP2 is much less than that of LP1, which may be related to the molecular weight of the polysaccharide. The apparent viscosity and solubility of the polysaccharides were associated with their molecular weight and glycosidic chain [27]. Generally, smaller molecular weights are typically related to increased solubility and decreased viscosity.

#### 3.2.2. Monosaccharide Compositions

The monosaccharide compositions of LPs were determined by GC-MS and the results are shown in Table 2. Both LPs contained rhamnose, arabinose, mannose, glucose and galactose, but the molar proportions of these monosaccharides were different, which revealed that the LPs were neutral heteropolysaccharides. This outcome was in line with our earlier reports on litchi pulp polysaccharide [6,30]. Arabinose and galactose accounted for more than 80% of the monosaccharide composition in LP2. In contrast, the amount of galactose in LP1 was essentially the same as in LP2, but the proportion of arabinose was much lower and the proportions of mannose and glucose were higher. Even though rhamnose was less abundant, the percentage of rhamnose in LP2 was obviously higher than that in LP1. In conclusion, the use of an ultrafiltration membrane may be responsible for the distinct difference in the monosaccharide compositions of the two fractions.

#### 3.2.3. FT-IR Analysis

The FT-IR spectrum of LP1 and LP2 exhibited polysaccharide characteristic absorption peaks (Figure 2). The hydrogen bonding was responsible for the absorptions in the range of 3700–3000 cm^−1^ and 1075–1010 cm^−1^, while the alkyl group was represented by the peaks between 3000–2800 cm^−1^ and at approximately 1410 cm^−1^. The absorption peak at 1370 cm^−1^ was attributed to the carbonyl C=O stretching vibration in uronic acid. The peaks at roughly 1143 cm^−1^, 1100 cm^−1^ and 1074 cm^−1^ could correspond to a characteristic C-O-C stretching vibration on the glucose-pyranose ring, while the absorption peak of 905–876 cm^−1^ represents β-D-pyranose. The FT-IR spectrum of LP1 and LP2 are similar, but the transmittance of some characteristic bands is different, which may be related to the sugar content. On the whole, the results confirmed that LP1 and LP2 were α-β-D-pyranose rings containing uronic acid.

### 3.3. Immunomodulatory Activity

Macrophages have many important functions, such as the phagocytosis of microbial pathogens and abnormal cells, and the production of cytokines. Cytokines control homeostasis by regulating cell differentiation, proliferation and apoptosis, as well as defense functions such as immune and inflammatory responses [28]. Therefore, we investigated the effects of LP1 and LP2 on the secretion of NO, IL-6 and TNF-α by macrophage RAW264.7; the results are shown in Figure 3. Although the abilities of LP1 and LP2 to stimulate NO release from macrophages changed little with concentration, the stimulating effect of LP2 was significantly stronger than that of LP1 at all concentrations. The NO production of macrophages stimulated by 100 μg/mL of LP1 and LP2 were 0.52 and 1.11 μmol/L, respectively, which were significantly lower than the positive control LPS (*p* < 0.05). In addition, both LP1 and LP2 promoted the production of TNF-α in a dose-dependent manner at the dose range of 50–100 μg/mL (Figure 3B). Compared with LP1, LP2 significantly enhanced the ability of macrophages to secrete TNF-α (*p* < 0.05) at all test concentrations. The LP2 medium group produced a higher amount of TNF-α than the LPS group at 75 and 100 μg/mL. Furthermore, the amount of IL-6 in the LP1 group was comparable to that in the blank group and did not vary as the concentration was raised; while the amount of IL-6 in the LP2 group was larger than that in the blank group and increased in a dose-dependent manner, showing that LP2 was more effective than LP1 at stimulating macrophages to secrete IL-6 (*p* < 0.05). The amount of IL-6 released by macrophages in the LP2 group at 50 μg/mL was less than that in the LPS group; however, at other concentrations, it was noticeably higher than that in the LPS group (*p* < 0.05).

As previous studies have found that litchi pulp polysaccharide exhibits immunomodulatory activity [2], we further investigated the immunostimulation of LP1 and LP2 obtained via ultrafiltration membrane treatment. According to the findings of our study, LP2 was more effective than LP1 at stimulating macrophages to secrete NO, IL-6, and TNF-α. The physicochemical differences between the two LPs could be the cause of this variation in activity.

The interaction of polysaccharides with immune cells through membrane receptors stimulates intracellular signaling cascades that have an impact on immunological responses [28,31]. Initially, the identification of polysaccharides by cell-surface receptors depends heavily on the composition of the monosaccharide. In this work, LP2, which had a higher ratio of arabinose and galactose than LP1, showed a larger capacity for immunomodulation. A study claiming that arabinogalactan is an active polysaccharide with immunomodulatory properties supported this finding [18]. Meanwhile, LP2, with its higher rhamnose content, was more immunoactive than LP1, with its higher glucose content, which was also present in longan pulp polysaccharide [32]. In the experimental concentration range, dried longan polysaccharides with rhamnose and galactose stimulated macrophage cytokine release substantially more than fresh longan polysaccharides with a predominant glucose content. Furthermore, molecular weight may also affect the expression of immunomodulatory activity of litchi polysaccharide. The ability of polysaccharides from *Porphyridium cruentum* [33] and *Abrus cantoniensis* [34] to modulate the immune system was strongly correlated with molecular weight, particularly low molecular weight. Our results are consistent with these previous researches. LP2 has a smaller molecular weight than LP1 and is more effective at stimulating macrophages to secrete NO, IL-6, and TNF-α than LP1. Finally, the viscosity and solubility of polysaccharide are also important factors affecting the immunomodulatory activity. This conclusion was confirmed in *Rhizoma panacis Japonic* polysaccharide [35]. In general, polysaccharides with low viscosity and high solubility are able to enhance fluid flow and ease the interface between polysaccharides and immune cells. This also explains why macrophages were more stimulated to secrete cytokines by LP2, which has lower viscosity and higher solubility than LP1. In conclusion, LP2 exhibited stronger immunomodulatory activity than LP1 as a result of the aforementioned structural properties, which makes it more valuable for development and research in the food and medical industries.

### 3.4. Prebiotic Activity

The stimulatory effects of LPs and FOS on the proliferation of *Bifidobacterium adolescentis*, *Lactobacillus casei*, *Leuconostoc mesenteroides* and *Lactobacillus plantarum* are shown in Figure 4. As for *Bifidobacterium adolescentis*, different carbon source media had significant differences on its proliferation (*p* < 0.05). FOS had the strongest prebiotic ability to promote the growth of *Bifidobacterium adolescentis*, followed by LP2 and LP1. The microbial populations were 4.56 ± 0.14, 3.45 ± 0.23, and 1.12 ± 0.05 log CFU/mL in the medium containing FOS, LP2 and LP1, respectively (Figure 4A). As demonstrated in Figure 4B, for *Lactobacillus casei*, the number of microorganisms in the LP2 medium was comparable to that of FOS, but almost four times that of LP1, indicating that LP2 had significantly stronger prebiotic activity than LP1 on *Lactobacillus casei* (*p* < 0.05). The increased bacterial populations of *Leuconostoc mesenteroides* stimulated with LPs and FOS is shown in Figure 4C. The proliferation effects of the three carbon source media were in the following order: LP2 > FOS > LP1. The number of probiotics in the LP1 medium was comparable to that in FOS, but significantly lower than that in LP2 (*p* < 0.05). As for *Lactobacillus plantarum* (Figure 4D), the proliferation effect of LP1 on bacteria is significantly weaker than that of FOS, but LP2 is higher than FOS, and the number of probiotics in LP2 is almost twice that in LP1. Overall, LP2 exhibited more beneficial effect on *Bifidobacterium adolescentis*, *Lactobacillus casei*, *Leuconostoc mesenteroides* and *Lactobacillus plantarum* proliferation than LP1.

Various probiotics secrete different enzymes, which affects how well they can utilize polysaccharides. In addition, numerous carbohydrate-activating enzymes are needed to depolymerize the backbone and side chains of plant polysaccharides because of their intricate and varied chemical structure [36,37]. Whether polysaccharides can be utilized by probiotics depends on their physicochemical characteristics. First, the molecular weight of the polysaccharide will affect its utilization by probiotics. The probiotics were able to make use of LP2 more readily than LP1 in this work, which may be because LP2 belongs to the smaller molecular weight of the ultrafiltration membrane interception, only 16.96 kDa. According to earlier researches [38,39], low-molecular-weight polysaccharides can encourage the growth of probiotics, which is consistent with our finding. Another important aspect that affects the prebiotic activity of polysaccharides is their monosaccharide composition. Compared to LP1, LP2 has a higher arabinose and galactose ratio and stronger prebiotic activity. This effect was demonstrated in the studies of Cano [40] and Macfarlane [41], where arabinose and galactose significantly outperformed other oligosaccharides in terms of prebiotic activity. This fully explains why LP2 is better used by probiotics than LP1.

In addition, the prebiotic activity of polysaccharide is also affected by viscosity and solubility. In the study of Zhang [42], polysaccharides with low viscosity were more utilized by gut microbiota. A different study on rapeseed polysaccharide found that RP1, which is more water-soluble than RP2, had a greater effect on probiotics [43]. This is in line with the findings of the present study, where LP2 was more utilized by probiotics than LP1 due to the higher solubility and lower viscosity of LP2. This could be because of the improved dispersibility of polysaccharide in solution and the ability to prolong the molecular chain, which provide more active sites for the carbohydrate enzymes released by the probiotics. Finally, it is easily used by probiotics, thereby, had better probiotic proliferative activity. It makes sense that LP2 has a more favorable impact on probiotics than LP1 due to its physicochemical characteristics. Beyond that, there may be more mechanisms of action to be explored in the future.

## 4. Conclusions

In this study, two litchi polysaccharide fractions with significantly different molecular weights were purified and prepared by an ultrafiltration membrane. The LPs exhibited significant differences in sugar content, monosaccharide composition, viscosity and solubility. The low molecular weight polysaccharide fraction (LP2) showed better in vitro immunomodulatory and prebiotic activities than the high molecular weight one. The superior biological activity of LP2 is related to its low molecular weight, high solubility, low viscosity, and rich arabinose and galactose composition. This result suggested that an ultrafiltration membrane could be applied to produce a highly-active litchi polysaccharide. In addition, it demonstrates the application potential of litchi polysaccharide in the food and pharmaceutical industries.

## Figures and Tables

**Figure 1 foods-12-00194-f001:**
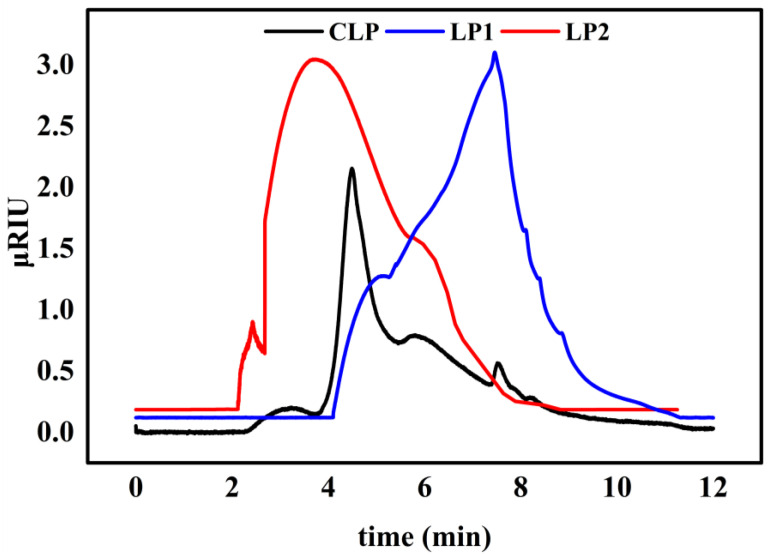
Chromatograms of the molecular weights of CLP, LP1, and LP2.

**Figure 2 foods-12-00194-f002:**
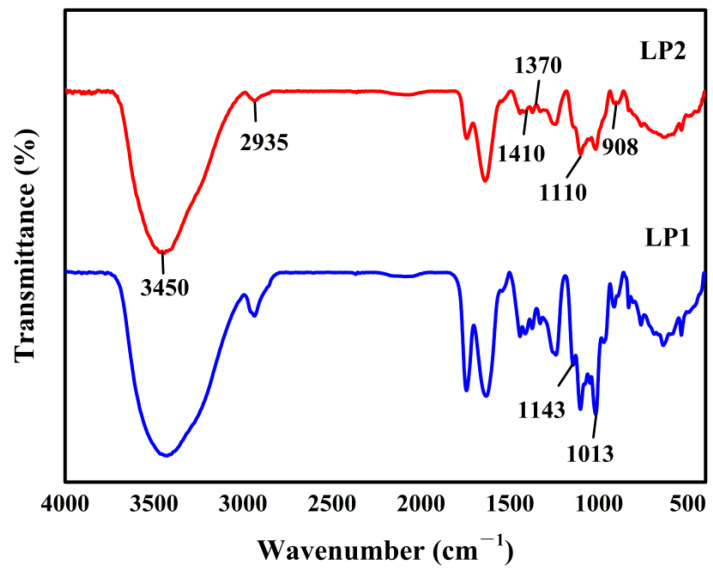
The FT-IR spectra of LP1 and LP2 were acquired over a frequency range of 4000–400 cm^−1^.

**Figure 3 foods-12-00194-f003:**
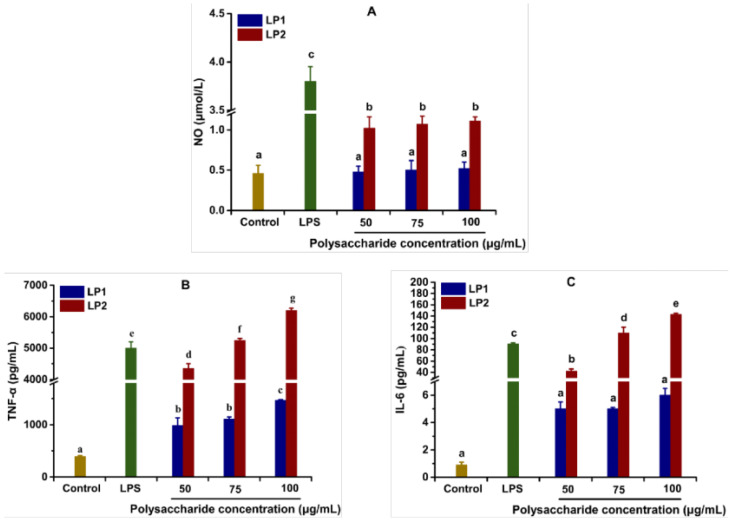
Stimulation effect of LP1 and LP2 on RAW264.7 macrophages. (**A**) NO, (**B**) TNF-α and (**C**) IL-6. Bars labeled with different letters represent a statistical difference at *p* < 0.05.

**Figure 4 foods-12-00194-f004:**
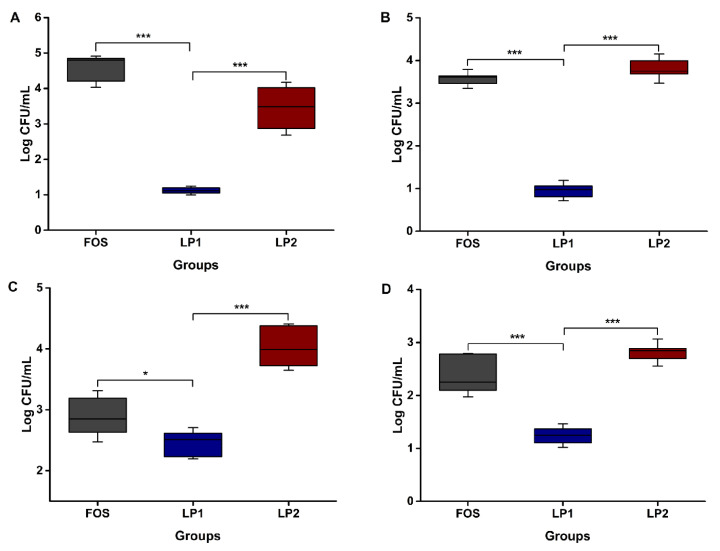
Stimulation effect of LP1 and LP2 on the population of *Bifidobacterium adolescentis* (**A**), *Lactobacillus casei* (**B**), *Leuconostoc mesenteroides* (**C**), and *Lactobacillus plantarum* (**D**) in a sugar free MRS base medium compared to FOS. * indicates a statistical difference of *p* < 0.05; ***, *p* < 0.001.

**Table 1 foods-12-00194-t001:** Basic characterization of LP1 and LP2.

	LP1	LP2
Yield (%)	67.33 ± 4.93	6.95 ± 3.95 *
Neutral sugar (%)	69.64 ± 2.26	78.63 ± 1.19 *
Uronic acid (%)	13.41 ± 2.30	8.99 ± 0.97 *
Protein (%)	2.02 ± 0.36	0.65 ± 0.01 *
Solubility (mg/mL)	14.37 ± 1.21	32.54 ± 1.97 *
Apparent viscosity (Pa∙s)	13.74 ± 0.28	2.57 ± 0.63 *

* indicates a statistical difference of *p* < 0.05.

**Table 2 foods-12-00194-t002:** The monosaccharide compositions of LP1 and LP2.

	LP1	LP2
Rhamnose (%)	0.90 ± 0.04	5.03 ± 0.12 *
Arabinose (%)	25.14 ± 0.17	50.73 ± 0.31 *
Mannose (%)	29.76 ± 0.12	7.58 ± 0.08 *
Glucose (%)	13.55 ± 0.13	4.49 ± 0.01 *
Galactose (%)	30.65 ± 0.21	32.17 ± 0.23 *

* indicates a statistical difference of *p* < 0.05.

## Data Availability

Data is contained within the article.

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
