# Peer review of "Purification, Characterization and Bioactivity of Different Molecular-Weight Fractions of Polysaccharide Extracted from Litchi Pulp"

_foods, 2023, doi:10.3390/foods12010194_

Round 1

Reviewer 1 Report

Dear Authors,

 After reviewing your manuscript, I think it may be major revisions. As can be observed by the authors, the indicated corrections are simple and should be considered as minor revisions. However, due to the number of these, the manuscript must be considered with major revisions. Please, refer to the revised PDF file to find the comments.

I ask you to carry out each of the indicated changes. After this, the manuscript can be accepted for publication in MDPI Foods.

Best regards

Reviewer 2 Report

Manuscript ID: foods-2096116

Title: Purification, characterization and bioactivity of different molecular-weight fractions of polysaccharide extracted from litchi pulp

The possibility of using an ultrafiltration membrane was tested to obtain active fraction of litchi polysaccharides.

 Although, the results are interesting, some details should be improved and explained. I would like to make some comments that authors could take into account to improve the overall quality of the manuscript.

Comments:

Line 115: The Sevag reagent was used to remove free proteins. Did you check the recovery of polysaccharides after this step? My experience shows that this method remove also significant fraction of polysaccharides. It was not mention in the manuscript how many mg of extract was received after water extraction per unit mass of raw material,  how many per cent was recovered after protein removal step.

Line 174: “Previous results” or preliminary results? Since the word “previous” requires appropriate citation of literature.

Line 190: The FOS were used but this material was not described; the source of FOS and average molecular mass of FOS should be given in the manuscript.

Line 191: The final concentration of LP1 and LP2 in MRS broth medium was 2%. It was not possible to prepare this medium with so high concentration of LP1 since LP1 solubility was 1.4% (Table 1).

Line 219: There are no relation between the solubility of polysaccharides and peaks response (LP1 and LP2). The concentration of solution to gel chromatography is usually not higher than 1-2 mg/ml (your polysaccharides were fully dissolved in this concentration). The key parameter for translating RI detector output to exact sample concentration is the dn/dc value, or refractive index increment.  This value is unique for a sample-solvent combination, as it represents the difference in refractive index between the sample and the solvent.

Table 1: The apparent viscosity has unit. (Pa ∙ s)

Line 184: the reference [28] is not justified here.

Round 2

Reviewer 2 Report

The paper has been corrected significantly and I think that final version of this paper can be considered for publication.